# Development and Validation of a Meta-Instrument for the Assessment of Functional Capacity, the Risk of Falls and Pressure Injuries in Adult Hospitalization Units (VALENF Instrument) (Part II)

**DOI:** 10.3390/ijerph20065003

**Published:** 2023-03-12

**Authors:** David Luna-Aleixos, Irene Llagostera-Reverter, Ximo Castelló-Benavent, Marta Aquilué-Ballarín, Gema Mecho-Montoliu, Águeda Cervera-Gasch, María Jesús Valero-Chillerón, Desirée Mena-Tudela, Laura Andreu-Pejó, Rafael Martínez-Gonzálbez, Víctor M. González-Chordá

**Affiliations:** 1Hospital Universitario de La Plana, Nursing Department, Universitat Jaume I, 12006 Castelló de la Plana, Spain; 2Nursing Research Group (GIENF Code 241), Nursing Department, Universitat Jaume I, 12006 Castelló de la Plana, Spain; 3Mathematics Department, Universitat Jaume I, 12006 Castelló de la Plana, Spain; 4Hospital Comarcal Universitario de Vinarós, Nursing Department, Universitat Jaume I, 12006 Castelló de la Plana, Spain; 5Hospital Universitario de La Plana, 12540 Villarreal, Spain; 6Nursing and Healthcare Research Unit (INVESTÉN-ISCIII), Institute of Health Carlos III, 28029 Madrid, Spain

**Keywords:** nurses, nursing, nursing assessment, hospitalization, validation study

## Abstract

The nursing assessment is the first step of the nursing process and fundamental to detecting patients’ care needs and at-risk situations. This article presents the psychometric properties of the VALENF Instrument, a recently developed meta-instrument with only seven items that integrates the assessment of functional capacity, risk of pressure injuries and risk of falls with a more parsimonious approach to nursing assessment in adult hospitalization units. A cross-sectional study based on recorded data in a sample of 1352 nursing assessments was conducted. Sociodemographic variables and assessments of the Barthel, Braden and Downton instruments were included at the time of admission through the electronic health history. Thus, the VALENF Instrument obtained high content validity (S-CVI = 0.961), construct validity (RMSEA = 0.072; TLI = 0.968) and internal consistency (Ω = 0.864). However, the inter-observer reliability results were not conclusive, with Kappa values ranging between 0.213 and 0.902 points. The VALENF Instrument has adequate psychometric properties (content validity, construct validity, internal consistency and inter-observer reliability) for assessing the level of functional capacity, risk of pressure injuries and risk of falls. Future studies are necessary to establish its diagnostic accuracy.

## 1. Introduction

Promoting the implementation of safe practices in patient care is one of the strategies proposed by the Ministry of Health in the last Patient Safety Strategy for the National Health System [1]. To this end, this document recommends an individualized care plan, which takes into account good care practices for the patient’s safety, such as the prevention of pressure injuries and falls, among others. However, Spain is one of the countries in the European Union with the highest costs derived from a lack of patient safety and adverse effects [2]. Specifically, there are significant safety problems with nursing care in Spanish hospitals, with a rate of 8.6% of pressure injuries, 3.6% falls [3] and patients with functional loss during hospital admission [4].

The occurrence of these adverse effects is linked to the failure in organizational aspects, working methods or tools, beyond individual errors [5]. In this sense, the assessment of care needs by nurses is the first step in detecting these risk situations that may translate into adverse events. To this end, nurses routinely use a wide variety of validated instruments, between eight and fifteen according to the literature, which may vary depending on the clinical context, units and hospitals [6]. The use of these instruments is essential for detecting real or potential problems in patients, although it increases the bureaucratic burden [7] and limits the direct care time [8]. Moreover, direct care is associated with a decrease in mortality [9], increased quality of care and user satisfaction [10], improved functional capacity of users and reduced pressure injuries and falls [11].

In another way, these instruments introduce a quantitative value through the assessment of specific and objective factors which, when considered as a whole, make it possible to classify patients according to the score obtained [12]. This ensures that nursing assessments are less subjective, increases the certainty of nurses and improves decision-making about patient care. [13]. However, these instruments are used independently, although they share constructs, dimensions and items related to mobility, hygiene or feeding, resulting in redundant assessments of care needs and problems associated with hospitalization [6,14]. This leads to skepticism among nurses and a perception of wasted time, making their acceptability and implementation difficult [15]. Consequently, nursing assessments become systematic and inaccurate, affecting their validity and thus the detection of patients at risk [6].

To avoid this, we carried out a research project that aims to collapse some of these instruments into a shorter meta-instrument more parsimonious with nursing assessment. In Part I [16], we present the development of the VALENF Instrument (by its Spanish acronym). Specifically, the VALENF Instrument collapses the Barthel (functional capacity) [17], Braden (pressure injury risk) [18] and Downton (fall risk) [19] instruments into a meta-instrument made up of seven items. The second part presents the psychometric properties (content validity, construct validity, internal consistency and inter-observer reliability) of the VALENF Instrument.

## 2. Materials and Methods

### 2.1. Design and Setting

A cross-sectional validation study based on recorded data was carried out in the Hospital La Plana in the Valencian Community (Villarreal, Spain). This was the reference hospital of one health department and covered around 200,000 inhabitants, according to data from the Municipal Register. The Ethics and Research Committee approved this study in December 2020 (code VALENF. 9 December 2020).

### 2.2. Participants and Sample

The target population comprised patients aged more than 18 years admitted to one of the seven adult hospitalization units in the participating hospital. Special services (intensive care, emergency, operating theatres or resuscitation), home hospitalization, maternal–infant and obstetrics’ hospitalization units did not form part of this study due to differences in the type of care processes, in the organizational model of these units or in the assessment instruments used.

The unit of analysis was nursing assessments. Thus, the study included nursing assessments of functional capacity (Barthel index), risk of pressure ulcers (Braden index) and risk of falls (Downton scale) in the first 24 h after admission to ensure that data related to the time of admission were obtained for all patients. Otherwise, the exclusion criteria were nursing assessments of patients transferred from other units at the same hospital or another hospital because their assessments when hospitalized did not correspond to the initial assessment.

The sample size was established based on the recommendations of the literature for the validation of instruments (between 5 and 10 participants per item) [20], although no specific recommendations about sample size were found when combining or unifying several instruments. In addition, the need to achieve a sufficiently representative sample and the need to work with subsamples in some phases of the analysis strategy were considered when including in the study all the nursing assessments carried out between September 2021 and January 2022.

### 2.3. Variables, Instruments and Data Collection

The VALENF Instrument is a more parsimonious solution for nursing assessment that allows for assessments of functional capacity, risk of pressure injuries and risk of falls in hospitalization units. The VALENF Instrument is a meta-instrument developed by the combination of seven items from Barthel (mobility), Braden (sensory perception, moisture and mobility) and Downton (previous falls, medication and sensory deficiency) indices [16]. It has a high predictive capacity regarding the global score of the Barthel (R^2^adj = 0.938), Braden (R^2^adj = 0.926) and Downton (R^2^adj = 0.921) indices. Moreover, it has high reliability with an intraclass correlation coefficient greater than 0.9 points [16].

Furthermore, the study included the variables of age, sex (male, female), process type (medicine, surgical), admission type (scheduled, emergency), main diagnoses according to International Classification of Diseases v-10 (ICD10) and the Charlson index for the study of comorbidity [21].

The nurses working in the included hospitalizations units carried out data collection as part of their normal work through the EHR between September 2021 and January 2022. In February 2022, the pseudonymized database was requested from the documentation service of the participating hospital, along with the variables to be studied, but without including any personal data that could identify patients. A consensus was reached beforehand with the documentation service regarding the structure of this database and this service kept the original database with patients’ identification details.

### 2.4. Validation and Data Analysis Procedures

Firstly, the VALENF Instrument’s content validity was determined. For this, a group of 15 experts was formed. This group included clinical nurses with at least 10 years’ experience, as well as university nursing teachers with a PhD degree and at least 10 years’ teaching experience in fundamental nursing or medical–surgical nursing. This group evaluated the suitability of the seven items to support nursing assessment as a global construct using an online questionnaire with a four-point Likert scale (where 1 represents “Nothing suitable” and 4 represents “Totally suitable”). Moreover, the group of experts assessed the suitability of the seven items to support functional capacity, risk of falls and risk of pressure injuries. To do so, we followed the methodology of Polit and Beck [22] and applied the Content Validity Index (CVI). The Content Validity Index of each item (I-CVI) was estimated by dividing the number of experts who scored that item as 3 or 4 points by the total number of experts. (suitable validity for items if I-CVI ≥ 0.78). The Global Content Validity Index (S-CVI) refers to the content validity of the instrument and was estimated as the mean score of the I-CVIs (suitable validity for instrument if S-CVI ≥ 9). One round was enough to reach adequate content validity.

Secondly, the VALENF Instrument’s construct validity was established, considering nursing assessment as a global construct. In addition, the construct validity of the items that were significant in Part I [16] to predict Barthel, Braden and Downton were studied. For this purpose, the sample was randomly divided into two subgroups and homogeneity between them was verified by an inferential analysis [23]. The Mann–Whitney U test (two groups) and Chi-squared test (categorical variables) were used, since it was previously confirmed that the data did not follow a normal distribution.

With subgroup 1 (*n* = 676), four explanatory factor analyses were run. Feasibility was confirmed by the Kaiser–Meyer–Olkin (KMO) test and by Bartlett’s test of sphericity. The oblique promax rotation method was used because moderate–high correlations were expected between the possible factors, and the principal axes method was used for factoring because the items did not have a normal distribution [24]. In addition, the Downton item scores were inverted so that all items measured in the same direction and factorial loadings over 0.3 indicated a good fit of the items [25]. Next, with subgroup 2, four confirmatory factor analyses were carried out using the maximum likelihood estimation technique. Goodness-of-fit was evaluated by means of the ratio of χ^2^ to degrees of freedom (χ^2^/df < 5 indicates an adequate fit), root mean square error of approximation (RMSEA, where <0.08 indicates a good fit), the comparative fit index (CFI ≥ 0.97 denotes a good fit) and the Tucker–Lewis index (TLI ≥ 0.97 denotes a good fit) [26].

Finally, the reliability was measured. On the one hand, the internal consistency was verified with McDonald’s omega (Ω > 0.7 indicated good internal consistency) [27]. On the other hand, two researchers piloted a sample of 41 patients to determine inter-observer reliability by the linear weighted kappa (poor agreement if *k* < 0.2; fair if *k* between 0.21–0.40; moderate if *k* between 0.41–0.60; good if *k* between 0.61–0.80; very good agreement if *k* > 0.80) [28]. The statistical analysis was performed with software JAMOVI V2.3.21 and MedCalc (20.218) for linear weighted kappa. Significance level was established at *p* < 0.05.

## 3. Results

### 3.1. Content Validity

Fifteen experts completed the content validity process. Of these, 66.66% (*n =* 10) were nurses with more than ten years of clinical experience and the rest were university nursing professors with a doctorate degree (3.33%; *n =* 5). The mean age was 44 (±9) years and only one expert was male. A single round was enough to reach adequate levels of global content validity (S-CVI ≥ 0.9). In addition, all items scored with an I-CVI higher than 0.78 when assessing their suitability to support nursing assessment as a global construct and their suitability to assess functional capacity and risk of pressure injuries. However, the Moisture item showed an I-CVI of 0.733 regarding the assessment of the risk of falls.

### 3.2. Construct Validity, Internal Consistency and Inter-Observer Reliability

#### 3.2.1. Descriptive Analysis of the Samples

We included 1352 nursing assessments that met the inclusion and exclusion criteria. The mean age of the sample was 67.69 (±17.92; Min = 18; Max = 101) years and 47.9% (647) of the nursing assessments were carried out on women. The mean Charlson index score was 3.68 (±2.5) points and 481 different main medical diagnoses were identified. Only 33.1% (*n =* 447) were surgical processes and 16.6% (*n =* 224) were scheduled admissions. Part I [16] shows a full description of the sample (Table 1).

Table 2 presents the descriptive and bivariate analysis of the randomized sample in two subgroups to study the construct validity. It can be observed that there were no significant differences in terms of age (*p* = 0.733) and distribution by sex (*p* = 0.956) or hospitalization units (*p* = 0.842). There were also no significant differences in the Barthel (*p* = 0.956), Braden (*p* = 0.826) or Downton (*p* = 0.895) scores, or in any of the other variables included in the study.

#### 3.2.2. Exploratory and Confirmatory Factorial Analysis

The results of the KMO test and the Bartlett sphericity test confirmed the viability of the four exploratory factor analyses shown in Table 3. On the one hand, the VALENF Instrument obtained a two-factor solution that explained 56.2% of the variance. The first factor (40.5% of the variance) grouped the items Mobility (Barthel), Sensory Perception, Moisture and Mobility (Braden), and the second factor (15.6% of the variance) grouped the items Sensory Deficiency, Previous Fall and Medication. The correlation between the two factors was *r* = 0.755 (*p* < 0.001).

On the other hand, three exploratory factor analyses are shown, in which only the VALENF items predicting Barthel, Braden and Downton, respectively, have been included. In this way, Barthel obtained a two-factor solution that explained 71.5% of the variance. The first factor (39.3% of the variance) included the items Mobility (Barthel) and Moisture and Mobility (Braden). The second factor (32.3% of the variance) grouped the items Sensory Perception (Braden) and Sensory Deficiency (Downton) with loads > 0.3 points. The correlation between the factors was *r* = 0.822 (*p* < 0.001). In addition, Braden obtained a one-dimensional solution that explained 62.1% of the variance, although the Medication item obtained a factor loading of 0.299. Finally, Downton obtained a two-factor solution that explained 52.2%. The first factor (36.1% of the variance) included the items Mobility (Barthel), Sensory Perception and Moisture. The second factor (16.1% of the variance) was composed of the items Sensory Deficiency, Previous Fall and Medication. The correlation between items was *r* = 0.742 (*p* < 0.001). The goodness-of-fit indicators were excellent in the four confirmatory factor analyses. Table 3 shows the complete results of the construct validity analysis. In addition, Appendix A includes the path diagrams of the four confirmatory factor analyzes.

### 3.3. Reliability

#### 3.3.1. Internal Consistency

The internal consistency of the VALENF Instrument was excellent, with a global Ω of 0.869 points for the seven items. However, the value of Ω increased slightly if the Sensory Deficiency and Previous Fall items were removed. In the same way, the internal consistency results were excellent when considering the items with predictive capacity for the Barthel, Braden and Downton instruments, although the increase in the Ω values when removing items related to the assessment of the risk of falls. Table 4 shows the complete results of the internal consistency analysis.

#### 3.3.2. Inter-Observer Reliability

Previous Fall was the only item that showed good–very good agreement (K = 0.905; 95% CI = 0.77–1) and Medication had a good reliability (K = 0.752; 95% CI 0.52–0.97). The items Mobility (Braden) (K= 0.605; 95% CI 0.44–0.76), Mobility (Barthel) (K 0.554; 95% CI 0.33–0.77) and Sensory Perception (K = 0.609; 95% CI 0.43–0.77) were moderate. Lastly, the Sensory Deficiency item reported poor reliability (K = 0.213; 95% CI −0.09–0.52). (Table 5).

## 4. Discussion

The VALENF Instrument was developed as a meta-instrument that combines other questionnaires used in nursing assessments [16]. In this manuscript, we present the first results of its psychometric properties (content validity, construct validity and reliability). Thus, the content validity was studied considering the nursing assessment as the global construct that measures VALENF Instrument, and, also, the content validity of the combination of its seven items considering functional capacity, risk of falls and risk of injuries due to pressure as constructs to measure. Content validity refers to the degree to which the items of an instrument represent the construct that it is intended to measure [29]. Content validity is considered the most important psychometric property since it allows one to specify whether all the content (items, tasks, observations, questions, etc.) of an instrument is relevant, complete and understandable with respect to the construct that the instrument measures [30]. As Palese et al. showed [16], the results of content validity were satisfactory and supported the fact that these instruments measure related constructs. However, Palese et al. [14] used two face-to-face meetings to establish content validity, while we used the Polit and Beck methodology [22], which added more rigor to the results. In addition, the Moisture item was the only item that did not obtain adequate content validity in relation to the assessment of the risk of falls, and this is consistent with the available evidence since the degree of exposure to moisture in any part of the body is not a risk of falls [31]. However, since the Moisture item obtained adequate I-CVI values considering the nursing assessment as a global construct, to assess functional capacity and to assess the risk of pressure injuries, the research team decided to retain this item.

The sample was divided into two groups to study the construct validity, as in the case of Palese et al. [14]. In addition, construct validity was analyzed separately for the VALENF Instrument and for the Barthel, Braden and Downton indices. On the one hand, the VALENF Instrument returned a structure with two factors that obtained excellent indicators of goodness of fit in the confirmatory factor analysis. The first factor grouped the original Barthel and Braden items into a dimension that we could define as Functional Capacity, although it also included the Moisture item, probably because it is an important risk factor in the development of pressure injuries. The second factor grouped the original Downton items that we could call Clinical Conditions. This could be a reflection of the correlation analysis carried out in Part I [16], where Braden and Barthel indices obtained a high overall correlation and a greater number of correlated items, while Downton’s overall correlation with both questionnaires was moderate and in fewer items.

These results partially coincide with those of Palese et al. [14], since their solution of 21 items was grouped into three factors. In their case, the first factor grouped eight items from the Barthel index and was called Functional Status. The second factor was called Cognitive Processes and grouped a total of ten items related to aspects as diverse as cognitive state, sensory perception, feeding or elimination. The third factor was called Clinical Conditions and grouped items related to medication, previous admissions or active medical problems. In this sense, it is worth mentioning that comorbidity, measured with the Charlson index [21], did not show a significant predictive capacity in the multivariate models created to develop the VALENF Instrument [16]. Moreover, Palese et al. [14] carried out their study with people older than 65 years in medical units. They included the Conley scale [32] instead of the Downton scale to assess the risk of falls and they also included the Blaylock Risk Assessment Screening Score (BRASS) [33] to assess the risk of prolonged hospital admission. These aspects may explain the differences in results.

On the other hand, the two-factor structure of the VALENF Instrument was partially replicated when studying the construct validity of the Barthel, Braden and Downton indices with the items that showed a significant predictive capacity for each of them in the multivariate models, performed to develop the VALENF Instrument [16]. Thus, the items that predict the Downton scale replicated the factorial model of the VALENF Instrument. The items that predict the Braden index returned one factor that coincided with the Functional Capacity dimension, although the Medication item obtained a factorial load below the established limit. In addition, it was grouped as a single item in a second factor, so it did not meet the criteria established by Goretzko to consider it as a dimension [24]. Finally, the factorial structure of the items that predict the Barthel index showed the most important differences, offering two factors with a new grouping of items. The first factor grouped the items related to Mobility and Humidity and the second factor grouped the items related to Sensory Perception. Despite these differences, the goodness-of-fit indicators and the internal consistency results were excellent for the four factorial models. However, it is striking how the McDonald’s Omega value increased if the original items of the Downton scale were removed in the internal consistency analysis. It is possible that this could be related to an inappropriate use of the Downton scale in Spain derived from errors in the translation of the original version [34], the lack of training and instructions on the use of this questionnaire or the difficulty of assessing the risk of falls through instruments that include risk factors [35].

As we explained in Part I [16], the Fundamentals of Care Framework [36] was used for the development of the VALENF Instrument. However, some authors [37] argue that the lack of conceptual clarity of the Fundamentals of Care hinders the development of measurement instruments, possibly because the disciplinary language of nursing still lacks consistency, as do the conceptual frameworks for to synthesize nursing actions in practice. In fact, we consider that the inter-observer reliability results are a reflection of this situation, since higher Kappa values could be expected as these are instruments that nurses use daily in their clinical practice, and that the items, a priori, can be considered of little difficulty. However, only the Medication item obtained a very good level of agreement, while the Mobility (Barthel index) and Sensory Perception (Downton scale) items obtained insufficient levels of agreement.

The limitations that must be taken into account when interpreting these results have already been largely included in Part I [16]. On the one hand, this is a retrospective study based on registry data from a single hospital and the sample selection was not random. On the other hand, the presence of cases with marginal scores (outliers), multivariate normality, covariance and collinearity, as well as the need to work with interval data are limitations inherent to the use of factor analysis that may have influenced the results of this study. Moreover, it is necessary to advance in the methodological consensus when developing meta-instruments that collapse other instruments, questionnaires, indices or tests. Even so, we believe that the results of this study are relevant to care management and nursing practice due to their clinical applicability. In fact, the VALENF Instrument is currently being implemented in the electronic clinical records of the hospital where it was developed. Due to the reliability results, interventions based on previous studies are being carried out, such as including help text on how to assess and interpret the items in the programming [38], setting up a team of assistants that will help in the implementation of the VALENF Instrument [39], carry out specific training [40] or schedule reassessment alerts [41]. Future studies should explore the opinion of nurses on the usefulness and applicability of the VALENF Instrument and other similar tools that may begin to be developed.

Urquarth et al. [42] conclude that there are various ways of conceptualizing nursing and articulating what is performed with patients, but nursing records’ systems, including nursing assessment, that are demonstrably effective, have not yet been developed. This may be because they are nursing record systems based on classical nursing theories grounded in paradigms from other disciplines [43]. Thus, the results of this study and others, such as that of Palese et al. [31], open lines of work related to the development of new nursing assessment instruments and, in addition, the possibility of advancing in the development of systems that allow the prescription of nursing care [44] and theory based on research results from a nursing perspective [43]. Thus, these results justify the need to advance in the construct validity analysis of the VALENF Instrument using techniques such as structural equation models and new samples. In addition, it is necessary to establish the diagnostic accuracy of the VALENF Instrument on the risk of pressure injuries, falls and functional capacity. Another aspect to consider is that the VALENF Instrument does not include items that allow for the assessment of care, such as feeding or elimination, despite the fact that they are aspects initially collected in questionnaires such as Barthel or Braden, and this opens the door to continue with their development, always taking into account the principles of parsimony and clinical applicability.

## 5. Conclusions

The VALENF Instrument has adequate content validity, construct validity and internal consistency as a meta-instrument capable of predicting functional capacity, risk of falls and risk of pressure injuries. However, it is necessary to advance in the analysis of its construct validity and in its development to include the assessment of other aspects, such as diet. In addition, the intra-observer reliability results justify the need to carry out actions that guarantee a correct implementation in the future.

## Figures and Tables

**Table 1 ijerph-20-05003-t001:** VALENF Instrument Content Validity Index.

Items	I-CVI ^1^			
	Nursing Assessment	Functional Capacity	Risk of Pressure Injuries	Risk of Falls
Mobility (Barthel)	1	1	0.93	1
Sensory Perception (Braden)	1	1	1	1
Moisture (Braden)	0.933	0.8	1	0.733
Mobility (Braden)	1	1	1	1
Sensory Deficiency (Downton)	0.933	0.933	0.8	0.933
Previous Fall (Downton)	0.933	0.933	0.8	1
Medication (Downton)	0.933	0.933	0.8	1
GLOBAL	0.961	0.942	0.904	0.952

^1^ I-CVI: Content Validity Index.

**Table 2 ijerph-20-05003-t002:** Descriptive and bivariate analysis of the subsamples.

Variable		Group 1 (*n =* 676)	Group 2 (*n =* 676)	
		m (ds) ^1^	m (ds) ^1^	*p* ^3^
Barthel index		78.40 (33.86)	78.36 (33.72)	0.956 *
Braden index		19.05 (3.82)	18.89 (3.92)	0.826 *
Downton scale		1.15 (1.23)	1.16 (1.23)	0.895 *
Age		67.8 (18.1)	67.6 (17.8)	0.773 *
Charlson index		1.31 (1.61)	1.32 (1.69)	0.608 *
		**% (*n*) ^2^**	**% (*n*) ^2^**	***p*** **^3^**
Sex	Male	52.07 (352)	52.22 (353)	0.956 **
	Female	47.93 (324)	47.78 (323)	
Process type	Medical	64.94 (439)	68.93 (466)	0.118 **
	Surgical	35.06 (237)	31.07 (210)	
Admission type	Scheduled	83.43 (564)	83.43 (564)	1.000 **
	Emergency	16.57 (112)	16.57 (112)	
Hospitalization unit	Traumatology	26.04 (176)	27.08 (183)	0.842 **
	Surgery and gynecology	19.82 (134)	21.59 (146)	
	Cardio/gastroenterology	14.64 (99)	14.05 (95)	
	Neuro/pulmonology	13.02 (88)	13.17 (89)	
	General surgery	2.66 (18)	1.92 (13)	
	Otolaryngology/urology	9.91 (67)	8.14 (55)	
	Internal medicine	13.91 (94)	14.05 (95)	

^1^ Mean (standard deviation); ^2^ Percentage (sample); ^3^ *p*-value; * Mann–Whitney U test; ** χ^2^.

**Table 3 ijerph-20-05003-t003:** Exploratory and confirmatory factor analysis.

Items	VALENF		Barthel		Braden	Downton	
	Factor 1	Factor 2	Factor 1	Factor 2	Factor 1	Factor 1	Factor 2
Mobility (Barthel)	0.944	0.069	0.943	−0.020	0.879	0.835	0.020
Sensory Perception (Braden)	0.689	0.219	0.171	0.769	0.862	0.735	0.184
Moisture (Braden)	0.818	0.045	0.478	0.408	0.858	0.892	−0.016
Mobility (Braden)	0.811	0.080	0.651	0.253	0.873	--	--
Sensory Deficiency (Downton)	0.034	0.365	0.009	0.631	--	−0.015	0.420
Previous Fall (Downton)	0.060	0.303	--	--	--	0.048	0.320
Medication (Downton)	−0.044	0.817	--	--	0.299	0.043	0.720
**Indicators of the exploratory factorial analysis**							
Bartlett’s Test	*p* < 0.001		*p* < 0.001		*p* < 0.001	*p* < 0.001	
^1^ KMO	0.885		0.875		0.860	0.832	
% Variance	40.5	15.6	39.3	32.3	62.1	36.1	16.1
% Total variance	56.2		71.5		62.1	52.2	
Correlation	0.755		0.822		--	0.742	
**Indicators of the confirmatory factorial analysis**							
^2^ χ^2^/df (*p*)	4.576 (<0.001)		2.55 (<0.001)		4.26 (<0.001)	3.8 (<0.001)	
^3^ CFI	0.980		0.997		0.991	0.985	
^4^ TLI	0.967		0.993		0.983	0.972	
^5^ RMSEA	0.072		0.048		0.069	0.064	

^1^ Kaiser–Meyer–Olkin test; ^2^ Ratio Chi-squared to degrees of freedom (*p*-value); ^3^ Comparative Fit Index; ^4^ Tucker–Lewis Index; ^5^ Root Mean Square Error of Approximation. The background color highlights the factors obtained.

**Table 4 ijerph-20-05003-t004:** Internal consistency.

	VALENF	Barthel	Braden	Downton
	Ω *			
Global	0.869	0.911	0.882	0.826
**if items are withdrawn**	Ω *			
Mobility (Barthel)	0.828	0.882	0.834	0.765
Sensory Perception (Braden)	0.826	0.880	0.838	0.753
Moisture (Braden)	0.830	0.884	0.839	0.762
Mobility (Braden)	0.826	0.881	0.835	
Sensory Deficiency (Downton)	0.884	0.924		0.803
Previous Fall (Downton)	0.886			0.845
Medication (Downton)	0.858		0.924	0.847

* McDonald’s Omega. The background color highlights the factors obtained.

**Table 5 ijerph-20-05003-t005:** Inter-observer reliability.

	Observer 2	Observer 1 (*n*; %)					K (95%CI) ^2^
		Independent	Needs Help	Wheelchair ^1^	Immobile	Total (*n*; %)	
Mobility (Barthel)	Independent	8	3	0	4	15 (36.6)	0.554 (0.33–0.77)
	Needs help	2	7	0	1	10 (24.4)	
	Wheelchair ^1^	0	0	0	0	0 (0)	
	Immobile	1	1	0	14	16 (39.0)	
	Total *n* (%)	11 (26.8)	11 (26.8)	0 (0)	19 (46.3)	41 (100)	
		**Completely limited**	**Very** **limited**	**Slightly** **limited**	**No** **impairment**	**Total**	**K** **(95%CI) ^2^**
Sensory Perception (Braden)	Completely limited	0	1	1	0	2 (4.9)	0.609 (0.43–0.77)
	Very limited	0	2	4	1	7 (17.1)	
	Slightly limited	0	0	8	4	12 (29.3)	
	No impairment	0	0	0	20	20 (48.8)	
	Total	0 (0)	3 (7.3)	13 (31.7)	25 (61)	41 (100)	
Mobility (Braden)	Completely limited	1	0	0	0	1 (2.4)	0.605 (0.44–0.76)
	Very limited	3	6	4	1	14 (34.1)	
	Slightly limited	0	2	5	5	12 (29.3)	
	No impairment	0	0	1	13	14 (34.1)	
	Total	4 (9.8)	8 (19.5)	10 (24.4)	19 (46.3)	41 (100)	
		**Constantly moist**	**Often moist**	**Occasionally moist**	**Barely moist**	**Total**	**K** **(95%CI) ^2^**
Moisture (Braden)	Constantly moist	0	1	1	0	2 (4.9)	0.609 (0.43–0.77)
	Often moist	0	2	4	1	7 (17.1)	
	Occasionally moist	0	0	8	4	12 (29.3)	
	Barely moist	0	0	0	20	20 (48.8)	
	Total	0 (0)	0 (7.3)	13 (31.7)	25 (61)	41 (100)	
		**No**	**Yes**	**---**	**---**	**Total**	**K (95%CI) ^2^**
Sensory Deficiency (Downton)	No	6	8	---	---	14 (34.1)	0.213 (−0.9–0.52)
	Yes	6	21	---	---	27 (65.9)	
	Total	12 (29.3)	29 (70.7)	---	---	41 (100)	
Previous Fall (Downton)	No	19	2	---	---	21 (51.3)	0.902 (0.77–1.00)
	Yes	0	20	---	---	20 (48.8)	
	Total	19 (46.3)	22 (53.7)	---	---	41 (100)	
Medication (Downton)	No	9	1	---	---	10 (24,4)	0.752 (0.52–0.97)
	Yes	3	28	---	---	31 (75.6)	
	Total	12 (29.3)	29 (70.7)	---	---	41 (100)	

^1^ Wheelchair-independent; ^2^ Linear weighted Kappa and 95% confidence interval.

## Data Availability

Data are available upon reasonable request. All necessary data are supplied and available in the manuscript; however, the corresponding author will provide the dataset upon request. All data relevant to the study are included in the article.

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
