# Peer review of "Development and Validation of a Meta-Instrument for the Assessment of Functional Capacity, the Risk of Falls and Pressure Injuries in Adult Hospitalization Units (VALENF Instrument) (Part II)"

_ijerph, 2023, doi:10.3390/ijerph20065003_

Round 1

Reviewer 1 Report

In my opinion, is relevant, necessary, and very well-crafted article. It uses clear, rigorous and thorough language.

A less positive aspect is the interdependence of reading the article - part 1. The method should also be more explicit that it is retrospective.

I would have liked to have read under the method that it had been authorized by the Ethics and Research Committee. Since it was only an option to place it in the "Institutional Review Board Statement"

Author Response

Dear reviewer, We attach a document to respond to your contributions.

Reviewer 2 Report

Methods:

Please shortly describe the selected services/hospitals en recruitment strategy. The descriptions in the text do not cover the need for information of the reader. The inclusions need to be discussed in the results section, preferably in a table.

"This study used recorded data collected between September 2021 and January 2022 through the electronic health record (HER). The documentation service provided the pseudonymized database in February 2022. " Please describe the methodology briefly, ie how was this done, by whom and with what means. This is necessary to know about the quality and nature of the data used in this study. 

Please elaborate on the expert panel round. How was CVI calculated for each item, what scale was used to score, was it online or live? 

Please check the verbs in this section - ie present vs past tense 

Results:

Much repetition of results section, please ficus methods on methods and results on results. Tables are sufficient to present the results.

Discussion:

The moisture item, why was it not discarded based on the results of the content validity? Please elaborate.

Your last sentence of the conclusion is very important: is this a limitation to the study, that implementation strategies were not considered, especially since you started the study to motivate nurses to use the instrument? How can correct implementation be established?

Please adjust the abstract to this conclusions, as it says inter-rater reliability is also high.

Author Response

(The authors gave the same response as above.)

Reviewer 3 Report

Thank you for the opportunity to review this article on validation which I found pretty interesting. Overall, this article may be of use clinically and to other researchers. However, some minor aspects could be revised to improve the overall quality of the text.

In Participants and sample, the authors write, «The target population, selection criteria, and sample size estimation are explained in detail in Part I.» I recommend moving this explanation, or at least the more general part, to this section, as it seems more natural.

In methods, the authors write, «Thus, we included 1352 nursing assessments that met the inclusion (carried out in 104 the first 24 hours after admission, assessments made of Barthel, Braden and Downton index) and exclusion (patients transferred from other units at the same hospital or from 106 another hospital) criteria. The mean age of the sample was 67.69 (±17.92; Min = 18; Max = 101) years, and 47.9% (647) of the nursing assessments were carried out on women. The mean Charlson index score was 3.68 (±2.5) points, and 481 main medical diagnoses were identified. Only 33.1% (n = 447) were surgical processes, and 16.6% (n = 224) were 110 scheduled admissions. Part I [16] shows a full description of the sample.» Here the authors show results. Thus, I would recommend moving these results to the results section.

The authors state that they performed exploratory and confirmatory analyses. Although the former is clearly explained in the text, in the case of the confirmatory analysis, the authors only show some of the many results that could be relevant, and that would indicate a good goodness of fit. It would be helpful to know whether these analyses were performed and whether the results were similar to those shown or whether they differed. These results could also be helpful for discussion or to reflect any potential limitations of the validation process.

In the case of having used software that allows the elaboration of graphs, it would also be helpful to display a graph showing the results of the confirmatory analysis, showing the relationship between the items and the domains established in the exploratory analysis.

Some parts of the text are written in the present tense, others in the future, and others in the past. I recommend reviewing this and applying the classic rule of writing introduction, methods, and results in the past tense and discussion in the present tense. It would also be helpful to review the use of English as there are some minor grammatical errors.

There are too many references to the first part of the study in almost every section of the paper. I understand that this could be acceptable, but it makes the reading of the article very complicated since it forces the reader to jump between both texts. The article's readability would be significantly improved by including at least a summary of the aspects of the first part that are continually mentioned. The readability of the article would improve considerably.

The section describing the limitations could be improved. The study design, the sample selection or inherent methodology, the content validation process, and the exploratory and confirmatory factor analyses have limitations that should be shown and help contextualize the results and their external validity.

Author Response

(The authors gave the same response as above.)
